# Happiness and Socio-Demographic Factors in an Italian Sample: A Propensity-Matched Study

**DOI:** 10.3390/healthcare11111557

**Published:** 2023-05-25

**Authors:** Matteo Rizzato, Michele Antonelli, Carlo Sam, Cinzia Di Dio, Davide Lazzeroni, Davide Donelli

**Affiliations:** 1Humandive, 33170 Pordenone, Italy; 2Department of Public Health, AUSL-IRCCS of Reggio Emilia, 42100 Reggio Emilia, Italy; 3Department of Psychology, Catholic University of the Sacred Heart, 20100 Milan, Italy; 4Prevention and Rehabilitation Unit, IRCCS Fondazione Don Gnocchi, 43100 Parma, Italy; 5Department of Medicine and Surgery, University of Parma, 43100 Parma, Italy; 6Division of Cardiology, Azienda Ospedaliero-Universitaria di Parma, 43100 Parma, Italy

**Keywords:** happiness, survey, Italian population, psychology, social sciences, quality of life, well-being

## Abstract

Happiness is receiving more and more interest both as a determinant of health and a measure of outcome in biomedical and psychological sciences. The main objective of this study was to assess how the levels of happiness vary in a large sample of Italian adults and to identify the socio-demographic conditions which impair happiness domains the most. The participants of this survey consisted of 1695 Italian adults (85.9% women; 14.1% men) who completed the Measure of Happiness (MH) questionnaire online. In this study, the differences between groups in total and single domain (life perspective, psychophysical status, socio-relational sphere, relational private sphere, and financial status) happiness levels were examined through a propensity score matching analysis with respect to socio-demographic conditions, including gender, age, annual income, relationship status, having children, and education level. The results show that low income has a negative impact on happiness levels, whereas being in a relationship has a positive effect. Having children appears to have a negative impact on male happiness. Males appear to be happier than females, especially with regard to the psychophysics status. This evidence emphasizes the urgency for Italian policymakers to take actions on removing obstacles to people’s happiness, especially with regard to financial distress, parenthood, and gender gaps.

## 1. Introduction

In the scientific community, happiness is receiving more and more interest as both a determinant of health and a measure of outcome in biomedical and psychological sciences [1,2,3]. Efforts have been made by researchers to provide a framework capable of integrating the diverse definitions of happiness, ultimately concerning how people evaluate their life and whether or not they are satisfied with it [4]. Happiness is not to be mistaken with joy, the latter being a short-lasting positive emotion, while the former being more like a background state of fulfillment in life than a temporary feeling of pleasure [3]. Therefore, happiness is deeply interconnected with well-being, which is in turn influenced by different factors, including psychophysical health, income, employment, social relationships, and life perspectives [5]. 

The concept of well-being and its relationships with happiness has undergone significant changes over time, and it is still a subject of discussion today. In the past, the absence of physical or psychological pathologies was regarded as well-being. However, in the 1960s, the concept of well-being shifted towards a balance between strengths and deficiencies in various domains of life, including the biological, psychological, social, and economic spheres. Positive psychology has emerged as a field that aims to identify the factors that enable people and communities to flourish and lead satisfying and meaningful lives [6].

There are two main perspectives regarding the interpretation of well-being: the hedonic and the eudaimonic perspectives [7]. The hedonic perspective equates well-being to happiness and pleasure [7]. It suggests that the meaning of life is to experience the greatest amount of pleasure derived from attaining any goal. The eudaimonic perspective, on the other hand, does not consider subjective needs and momentary pleasures but focuses on those needs that are part of human nature and lead to growth [8]. The eudaimonic perspective emphasizes the importance of an ideal toward which one strives, giving meaning and direction to one’s life [9,10,11,12]. Some argue that the constructs of “hedonia” and “eudaimonia” are both overlapping and distinct [13]. Both pursuits equally relate to vitality, and both show some links with life satisfaction, although hedonia’s links are more frequent.

Following the dichotomic nature of well-being, happiness is influenced by intrinsic and extrinsic factors too. The “Goal Contents Theory” suggests that striving for extrinsic motives, such as fame, possessions, and money, can lead to low levels of accomplishment and job/life satisfaction [14]. In contrast, happiness may find roots in a sense of accomplishment and intrinsic trust [15]. Additionally, the concept of happiness should not be mistaken with that of “contentment”, the latter being more relative than the former; in other words, happiness does not depend on situational factors or comparisons, but on the overall evaluation of different life domains [16].

Education has been viewed as a key factor in promoting individual happiness, while also driving national prosperity and social welfare [17,18]. Education can also help individuals develop important skills in areas such as time management and relationship building, leading to higher levels of satisfaction and happiness in relationship-based life domains [19]. The relationship between education and happiness seems bidirectional, as higher education is usually associated with more life satisfaction, and, in turn, psychophysical well-being facilitates good study achievements [20,21].

Personal growth and life perspective are crucial factors in enhancing one’s overall happiness, mostly through the enhancement of cognitive happiness, which is closely linked with the eudaimonic perspective and describes the integration of meaning, engagement, and pleasure [22]. Engagement is a psychological construct that involves how individuals are invested in their activities at a physical, emotional, and cognitive level, and is one of the factors influencing cognitive happiness the most [23]. 

Financial wealth is also an important determinant of personal growth, and plays a vital role in enhancing happiness and well-being [24], as it helps achieve better standards of living through access to more resources and opportunities [25]. In line with these general observations, previous investigations in Italy have found that income and wealth increase happiness and that unemployment is extremely harmful to subjective well-being; however, income received from public social transfers has a limited impact on happiness [26].

Although several studies did not find a specific correlation between happiness and age, recent evidence suggests that individuals gradually become less happy after reaching the age of eighteen, with a minimum point observed around the age of fifty, followed by a gradual increase in happiness as they get older, and then again a decline in very old age (some authors describe it as the “U-shaped” relationship between happiness and age) [27,28]. The reasons for this phenomenon are still unclear, but some hypotheses suggest that diverse factors, typical of different stages of life (such as income, job-related issues, relationships, longevity, and self-rated health) may play a role in it [27,28].

The sphere of interpersonal relationships has also been considered in relation to happiness, and the hypothesis is that a more intense relational life has a greater impact on the happiness of women, elderly individuals, and those with less education [29]. Moreover, it has been demonstrated that, beyond sentimental relationships, social capital has a significant impact on an individual’s well-being, regardless of other factors, suggesting that network support and the quality of interactions within a community play a key role in influencing the levels of happiness [30,31]. The positive relationship between the quality of relationships and subjective happiness is widely evidenced, including in Italian studies [32,33]. 

The role of happiness has even been investigated in relation to parenthood, as well as to the impact of having children on both male and female parents. Becoming a parent might thus be linked to lower initial levels of well-being (e.g., due to parenting stress, economic strains, etc.), but may be an investment in future well-being and insurance against loneliness and lack of support later in life [34]. Recent research has further explored the effect of childbirth on parenthood, with a focus on the quality of life in the early years after the birth of children [35].

Considering the key role of happiness in the individual’s well-being [36], workplace productivity [37], policy-making [38], and ultimately community life and public health [39], psychology research has tried to find strategies to objectively measure happiness through validated questionnaires. Most of these assessment tools are psychometric tests aimed to explore different well-being-related life domains, adapted on the basis of the population sampled (i.e., children, adolescents, adults, workers, people with markedly diverse socio-cultural background, etc.) [40,41,42]. In fact, being able to quantify happiness and to understand which domains are mostly involved in life dissatisfaction and impaired well-being can help policymakers, employers, and even single individuals to adopt proper strategies and take action in the right direction.

The pursuit of happiness has also been involved in the foundation of entire countries, with the US Declaration of Independence reporting it as a citizen’s right [43] and the Bhutanese government creating the so-called “Gross National Happiness”, an index aimed to measure and promote happiness across the population [44]. For the same reasons, every year, the United Nations (UN) publishes the World Happiness Report, a list where all countries are rated on the basis of a combination of parameters such as GDP per capita, social trust and support, health-related quality of life, individual freedom, and perceptions of corruption [45]. In the past few years, the top countries were mostly those of the northern part of Europe, with Finland being ranked as the best for several years consecutively [46]. On the contrary, Italy has never scored better than the 30th position in the past few years, falling behind the most developed Western countries in this international ranking of domestic happiness [45]. The reasons for this result probably lie in a combination of socio-economic, cultural, health-related, demographic, and political issues that the country is struggling to effectively tackle. In general, the specific determinants of happiness have been investigated only partially in Italy. In a study conducted in 2014, it was demonstrated that there is a strong correlation between happiness and self-perception of health in the northern part of the country (the better self-rated health was, the higher were the levels of happiness) [47]. Other studies conducted in Italy showed that the levels of happiness tend to be lower in middle-aged subjects (55–59 years old) if compared with younger responders [48], and that both economic and non-economic (social capital-, education-, urban health-, and employment-related) factors play a fundamental role in contributing to subjective well-being [26,49,50]. However, further investigations would be useful to better study happiness among the Italians, from both a qualitative and a quantitative point of view. 

The main objective of this study was to assess how the levels of happiness vary in a large sample of Italian adults and to identify the socio-demographic conditions which impair happiness domains the most.

## 2. Methods

### 2.1. Participants and Study Design

The participants of this study consisted of 1695 Italian adults. The inclusion criteria for participating in the survey were to be over 18 years old and to be an Italian fluent speaker. The study was conducted during 2022 in the context of the validation of the MH questionnaire. The survey was advertised through social media. In these sources, those who were interested in taking the test could find a link to an online registration form and, upon completion, they were allowed access to a personal dashboard with the questions to complete. The Measure of Happiness (MH) questionnaire [3] was administered online through a dedicated digital platform, utilizing a custom algorithm that randomized the order of the questions and answer options in order to prevent any repetitive behavior. Participants could choose to log out and resume the test within two hours. To ensure confidentiality, data were securely sent through a Secure Socket Layer (SSL). The participants were provided with all necessary information, including the purpose, instructions, and duration of the survey, and were informed that they could exit the session at any time and resume it later. This was carried out to ensure that, on the one hand, all data were collected at approximately the same time point and, on the other hand, that respondents were encouraged to complete the whole test. Prior to the administration of the questionnaire, all participants were required to provide an informed consent and were treated in accordance with the ethical guidelines outlined in the Declaration of Helsinki and its subsequent revisions. The study was approved by the local ethics committee of “Università Cattolica del Sacro Cuore”, Milan (Italy). After the collection of demographic information, the participants were asked to complete the MH questionnaire. 

### 2.2. Measure of Happiness Questionnaire

The MH is a self-report psychometric test developed to assess the construct of happiness through questions addressing several aspects of the individual’s life [3]. The questionnaire consists of 14 items grouped into five domains: Psychophysics Status (3 items), Financial Status (3 items), Relational Private Sphere (3 items), Socio-Relational Sphere (2 items), and Life Perspective (3 items). Each item is rated on a 10-point Likert scale ranging from 1 (not at all) to 10 (very, very much), and the scores obtained are used to estimate the levels of both overall and domain-specific happiness. The 14 items (in Italian and English) of the MH are the following:

Psychophysics Status:○Come valuti il rapporto con il tuo corpo?■How do you evaluate your relationship with your body?○Come valuti il tuo livello di equilibrio mentale e fisico?■How do you evaluate your level of mental and physical balance?○Come valuti il tuo rapporto con te stesso?■How do you evaluate your relationship with yourself?

Financial Status:○Quanto ritieni di essere realizzato in questo momento?■How fulfilled do you feel with your life at this moment?○Quanto sei soddisfatto della tua condizione finanziaria?■How satisfied are you with your financial situation?○Quanto ti senti solido finanziariamente?■How financially sound do you feel?

Relational Private Sphere:○Come valuti la qualità dei tuoi rapporti con i tuoi affetti principali?■How do you evaluate the quality of your relationships with your dear ones?○Quanto ti soddisfa l’atmosfera che si vive nella tua attuale casa?■At present, how satisfied are you with the atmosphere in your home?○Secondo te, i membri della tua famiglia, quanto ti stimano?■In your opinion, how much do your family members appreciate you?

Socio-Relational Sphere:○Quanto pensi che le persone, in generale, siano felici di relazionarsi con te?■In general, how happy do you think people are to interact with you?○Quanto ritieni apprezzati i tuoi comportamenti nella società?■How much do you think your behavior is appreciated in society?

Life Perspective:○Quanto ritieni importante porti degli obiettivi di lungo termine?■How important is it to you to set long-term goals?○Quanto ti interessi al tuo miglioramento personale?■How much are you engaged in self-improvement?○Quanto ti senti flessibile di fronte ai cambiamenti della vita?■How adaptable do you feel to major changes in your life?

A secondary objective of the present study is to assess the performance of the MH questionnaire. Further details about this psychometric test can be found in the validation study published by Rizzato et al. in 2022 [3]. 

### 2.3. Data Analysis

The statistical analysis was conducted with the “R” software (R Development Core Team 2020) and RStudio ver. 2022.07.2 [51], using the “matchit” [52], “marginaleffects”, and “stats” packages [53]. Microsoft Excel was used to organize the collected data. The threshold for statistical significance of the overall effect size was set at *p* < 0.05.

In this study, the differences between groups in total and single domain (life perspective, psychophysics status, socio-relational sphere, relational private sphere, and financial status) happiness levels were examined with respect to several demographic variables, including gender, age, annual income, relationship status, having children, and education level. A gender-based subgroup analysis was performed to better investigate any significant differences in male and female participants considered separately.

The propensity score matching was utilized to estimate the average marginal effect on total and single domain happiness levels, while controlling for confounding by the available baseline characteristics, of belonging to the studied categories, which included gender (male versus female), having an annual income below (or above) EUR 21,000, having (or not having) children, having a higher education (university degree) versus having a lower level of education, being engaged versus being single, and being under (or above) 30 years old [52,54,55,56,57,58]. No participants with missing data in any of these covariates were present; therefore, no participants were excluded from the final analysis. Baseline covariates were considered balanced between the groups in the propensity-score-matched sample if the absolute difference of the standardized difference was ≤0.15, and considering <0.1 as optimal balancing. Initially, several trials with nearest neighbor propensity score matching with and without replacement and with various 1:k ratios were attempted, but they did not yield satisfactory balance. Therefore, a full matching method on the propensity score was tried, which yielded a satisfactory balance. The propensity score was estimated using a logistic or a probit regression of the index condition (belonging to one of the two groups) on the covariates, choosing the one yielding better covariate balance. When only suboptimal covariate balance was achievable, spline function was applied on covariates to obtain satisfactory balance [59].

After matching, all standardized mean differences (SMDs) for the covariates were below 0.1 or 0.15, thus indicating an adequate balance (Figure 1). The only exception was being single versus engaged in a relationship in the sample of male participants, for which the best balance yielded was an SMD of <0.2 and, therefore, this condition was not studied. Since full matching uses all units allocated to both the studied condition and the control group, no units were discarded.

The efficacy endpoint, which was any significant difference in total or domain happiness due to the possess of the studied condition, was compared across the propensity-matched groups fitting a linear regression model: the total MH score along with life perspective, psychophysics status, socio-relational sphere, relational private sphere, and financial status subscores were the outcomes of interest, and the studied condition, other covariates, and their interactions were considered as predictors. The analysis also included the full matching weights to estimate the intervention effect and its confidence interval. The g-computation method is a statistical technique used to estimate the causal effect of an exposure or intervention on an outcome in the presence of confounding factors. In this study, the g-computation method was used to estimate the statistic known as the average treatment effect on the treated (ATT) in the propensity-score-matched sample, which in our case represents the average effect in terms of total or domain happiness on the individual having the condition of interest. The cluster–robust variance was used to estimate the confidence interval for the ATT with the matching stratum membership as the clustering variable.

For statistical power analysis, a clinically meaningful difference likelihood analysis was performed using 95% CIs for means along with effect sizes and *p* values to draw better conclusions from the results [60].

## 3. Results

The study participants were 1695 Italian adults: 85.9% of them were women with a mean age of 34.5 years old (SD = 10.7), while the remaining 14.1% were men with a mean age of 40.0 years old (SD = 13.3). The most relevant sociodemographic characteristics of the study population are reported in Table 1.

There were participants from any Italian region, but the majority of respondents were from the northern part of the country (*n* = 748 out of 1695, or 44.1% of the overall sample), while those from central (*n* = 490, or 28.9% of the sample) or southern regions (*n* = 457, or 27.0% of the sample) were less represented (Figure 2). In the period 2019–2021, the demographic characteristics of the Italian population could be summarized as follows: Italians were around 59 million (males: 29 million; females: 30 million), their mean age was approximately 45.9 years old (ranging from 43.3 to 49.2, depending on the specific region), people with a university level of education varied from 8.1% to 27.8% depending on the age range, those who were employed accounted for 38.1% of the entire population (average annual salary: ~EUR 28–29 K), people engaged in a marriage were slightly less than half of the adult citizens (48.3% of men and 46.3% of women), and urban dwellers comprised the majority (~71%) of all residents [61].

Map created with https://www.mapchart.net/italy.html (access date: 16 April 2023).

The most important results of the propensity-score-matched analyses are reported in Table 2. In particular, males showed significantly higher levels of total happiness (ATT = +3.2 MH score; *p* < 0.05), while people with an annual income below EUR 21,000 (regardless of their gender) were less happy than those with a higher salary (ATT = −7.2 MH score; *p* < 0.05). In particular, males had significantly higher levels of happiness in the Psychophysics and Financial Status domains; having a low income was associated with significantly lower levels of happiness in almost all domains (see Table 2). Having children in Italy does not seem to be associated with significantly higher or lower levels of total happiness (ATT = −2.0 MH score; *p* = 0.13). The same can be said when comparing subjects with higher versus lower education (ATT = −0.3 MH score; *p* = 0.79), even if those with higher education had slightly but significantly better scores in the Life Perspective domain (and lower levels of happiness in the Relational Private Sphere domain). Being engaged in a relationship seemed to be associated with higher levels of happiness if compared with being single (ATT = +3.0 MH score; *p* < 0.05). Younger age (<30 years old) did not appear to have a significant impact on the individual’s levels of total happiness (ATT = −1.0 MH score; *p* = 0.44). However, younger individuals showed significantly lower levels of satisfaction in the Financial Status domain.

The subanalysis of data from female participants suggested that having a low income significantly impairs the levels of overall happiness (ATT = −7.3 MH score; *p* < 0.05), while the other variables were not associated with significant differences in total happiness scores. For males, both earning a low income (ATT = −6.5 MH score; *p* < 0.05) and having children (ATT = −5.7 MH score; *p* < 0.05) were associated with significantly lower levels of total happiness. In particular, females with a poor income had all their happiness domains significantly reduced, while males with a low salary had mostly two happiness domains significantly compromised: the Relational Private Sphere and the Financial Status. Males with children reported significantly lower levels of happiness in the Socio-Relational Sphere and in their Psychophysics and Financial Status, while females with children reported higher scores in the Life Perspective domain. The results for each specific happiness domain are reported in Table 2. 

## 4. Discussion

Some considerations can be drawn after briefly analyzing the main findings of this study, which was aimed at understanding how the levels of happiness can vary in a large sample of Italian adults. Overall, females, those with a low income, and people not engaged in a relationship appear to be less happy than their counterparts (males, individuals with a higher income, and married subjects). Education, age, and having children do not seem to significantly influence the levels of overall happiness but can modify the scores of specific domains.

In particular, young age (<30 years old) is associated with lower levels of financial happiness, but it does not seem to influence the other domains of happiness in a significant way, although it cannot be excluded that, with a larger sample, an effect might be detected. Moreover, being a woman under 30 years old shows a specific negative impact on self-perception and social relationships. These results are compatible with data from other Italian surveys, where it was demonstrated that the number of those aged 15–29 years old and involved neither in working activities, nor in education or training programs (called “NEETs”) has increased from 19.6% in 2004 to 25.1% in 2020 [62]. These young people, now forming a relatively greater proportion of the overall Italian population, are reported to face more and more difficulties with finding social and, in particular, institutional support to guide their transition into the labor market [63]. These findings are different from data collected in other Western countries, in which age does not appear to influence financial happiness, and where the young tend to have a more positive perception of their quality of life [64,65]. Overall, this indicates that young people need more support to properly integrate themselves into Italian society and to become a happier and more active part of it. 

Our analysis suggests that males tend to have higher levels of overall happiness if compared with females. The domains where male subjects show higher levels of happiness are those associated with their psychophysics and financial status and this is consistent with previous studies [66].

Although the number of males enrolled in the study was limited if compared with that of females, power analysis showed a high likelihood of a clinically meaningful difference especially in the psychophysics and financial status happiness domains, while the life perspective, socio-relational sphere, and relational private sphere happiness domains are unlikely to be affected by gender. All the same, data from the psychological literature seem to confirm the fact that, despite recent progress in emancipation of women, female adults living in Western countries tend to report lower levels of subjective well-being than in the past [67]. Country-specific socio-cultural characteristics can play a role in this phenomenon, and, even though women have a lower income and worse living conditions on average, they tend to be more optimistic than men (in other words, if they had the same conditions, opportunities, and resources as men, they would rate themselves as happier than their male counterparts) [68]. Regarding the condition of women under the age of 30, the data suggest that happiness significantly decreases in the sphere of the psychophysics status happiness domain. This area pertains to the relationship that women have with themselves and presents questions regarding self-esteem, as indicated by previous authors, which defines global self-esteem as the positivity of one’s overall regard for oneself as a person [66]. This gender gap is also aligned with the studies of Zuckerman et al., which highlighted that at every stage of life, there is a difference in self-perception in favor of males, both in terms of global and specific self-esteem [69,70,71]. So, taken together, these observations suggest that gender-gap-related issues are far from being solved and should be a priority in the policymakers’ agenda. 

Higher education seems to have no impact on overall happiness, but it can guarantee higher levels of domain-specific happiness through a better ability and commitment to self-improvement. This may be explained by the fact that education offers better tools and strategies to pursue personal development [72]. On the contrary, higher education is associated with lower levels of happiness in the domain of private relationships. Potential explanations for this finding may be the fact that study- and work-related issues are more often a priority amongst higher-educated people [73], who may have higher expectations in private relationships, thus being more selective with friends and partners. 

Parenthood is not overall associated with levels of happiness significantly different from those who do not have children. In particular, from results on the entire sample, it appears unlikely that having children significantly influences the life perspective and socio-relational happiness domains. However, interesting results emerge from the gender-based subanalysis. Having children has a negative effect on men’s happiness, involving financial, psychophysical, and social-relational domains. The main impact of having children on men’s happiness seems to be represented by the financial sphere: this suggests an urgent intervention by Institutions and politics to support parenting, through interventions directly targeted to the weakest income brackets. The presence of financial difficulties, combined with parenting, are factors of instability in family relationships, as previous studies have found [74,75]. This is in contrast with common beliefs, which suggest that having a child has a positive impact on the parents’ well-being, even though having more than one or two children does not increase this beneficial effect [76]; additionally, as people age, having children seems to have a more marginal influence on the individual’s happiness [77]. These data contribute to a debate already addressed in previous research on the complex topic of parenthood in relation to overall well-being. Prior studies have identified financial concerns as a disruptive factor to well-being in parenthood [78]. However, in a review of psychological research, it was concluded that, despite common beliefs, having a child is not automatically associated with higher levels of happiness, and some factors commonly responsible for problematic parenthood include low socio-economic status, poor education, being a single parent, and living in a non-pronatalist country [79]. As the Italian population is aging, facing a staggering economy and a declining welfare system [80,81,82], it may be unfortunately easier for new parents to be concerned about the future of their family and, therefore, to be less happy and satisfied with their life if compared with how they are expected to be. In Italy, family-oriented policies are urgently needed to help those raising children and the educational system in order to slow the Italian demographic and psychosocial decline, which is likely to have a very negative impact on the entire society in the following years, if this social issue is not adequately tackled [83,84]. For females, having children seems to be beneficial in terms of the life perspective happiness domain.

Having a low income (lower than 21.000 EUR/year in Italy) implies a markedly negative impact on overall and domain-specific (psychophysics and financial status, relational private and socio-relational spheres) happiness, except for that associated with self-improvement and resilience. This finding is consistent with recent evidence from other studies [24]. Even though it is commonly said that “money does not buy happiness”, the main findings of this survey indicate that being poor is associated with lower levels of happiness in a large sample of Italian people. In fact, recent research shows that money and, even more, time can increase happiness, and that spending money on shared experiences can boost happiness the most [85]. So, with very limited financial resources at one’s disposal, one’s life can be more miserable and less satisfactory, and a negative impact on the individual’s happiness is likely to be observed. Our observations confirm the fact that in the presence of low income, the overall level of happiness decreases, negatively influencing the private sphere of intimate relationships [86]. The low level of income, and generally low economic satisfaction, can be considered a predictive factor for the stability or instability of a relationship [87]: again, these findings imply the need for targeted policy actions to support families and parenting, especially among those with lower incomes.

Individuals who are in a committed relationship with a partner or spouse tend to experience greater levels of happiness than those who are single, with the most significant impacts being observed in their private relational and financial domains. These findings are consistent with a vast body of psychosocial research which suggests that being married or involved in a romantic relationship can enhance one’s overall happiness, and this effect persists even when comparing individuals in long-term relationships with those who are single [88]. These observations highlight the crucial role that intimate relationships play in shaping an individual’s happiness and overall well-being.

Finally, this study has demonstrated that the MH questionnaire is a reliable tool for assessing happiness across various domains, with findings consistent with the existing literature.

### Limitations

This study was based on a cross-sectional survey of adult responders, mostly females, and therefore characterized by an unbalanced male-to-female ratio. When interpreting our findings, it is important to account for the fact that the study involved more female than male participants. To obtain a more complete and accurate representation of Italian society, where the number of males and females is almost equal (the national sex ratio is estimated to be around 0.96 [88]), future surveys should aim to include a larger sample size of individuals of all genders. Despite the numerical differences between the male and female groups, a post hoc statistical power analysis showed that the male sample still had a 99.4% power to detect a difference between two different groups with an alpha set at 0.05. Moreover, the propensity-matched analysis was used to ensure that males and females were matched by similar characteristics, although with a ratio greater than 1:1.

Additionally, the sample was exclusively made of Italian people, whose social, cultural, and economic characteristics may differ from those of responders from other countries of the world. No data about children or adolescents’ happiness were collected because such a research effort would have required a different approach and other psychometric tests, specifically validated for that age range.

## 5. Conclusions

The emerging finding is that having a relatively low income in Italy has a severe impact on happiness. Therefore, in addition to structural interventions in the labor and professional markets, cultural and psychological support interventions are desirable, especially for young people, to revise their relationship with the value of money. A society that equates happiness with the possession of money creates an escalation that can negatively affect overall happiness. 

However, policymakers should also consider these findings as a necessity to promote socio-economic strategies aimed at easing a full and more efficient integration of younger generations in the labor market (for example, lower employment and income taxes for junior workers, simplified bureaucracy for young entrepreneurs, and better financial inclusion with easier access to credit), so that young people can earn a higher income and achieve a better quality of life.

Being in a relationship has a positive effect on happiness compared to not being in one, provided there are no financial problems. Therefore, training and awareness interventions are suggested towards a culture of family as a source of increased well-being. 

It may be advisable to investigate in depth the psychological and social reasons why having children represents a decrease in happiness among the Italian male population. Apart from investigating the psychosocial determinants of this observation and considering the steep demographic decline of the Italian population, our findings urge the need for more family-oriented policies, including lower levels of taxation, parental support at work, and free education from the nursery school for those who decide to have children. 

A concerning finding of this study is that men are happier than women, and this is likely in line with previous research on the difference in self-esteem and self-perception, favoring males. This evidence emphasizes the urgency of political actions aimed at reducing this gender gap and examining whether it is due to a lack of equal opportunities or the presence of a culture that penalizes being a woman in Italian society. 

## Figures and Tables

**Figure 1 healthcare-11-01557-f001:**
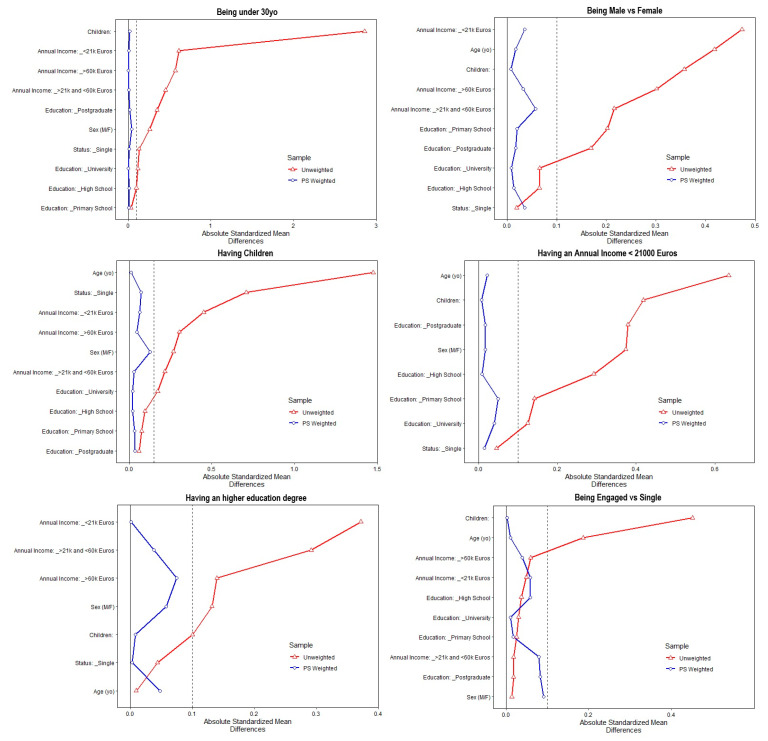
The Love plots show the balance of covariates before and after propensity matching for the condition of interest on the entire sample.

**Figure 2 healthcare-11-01557-f002:**
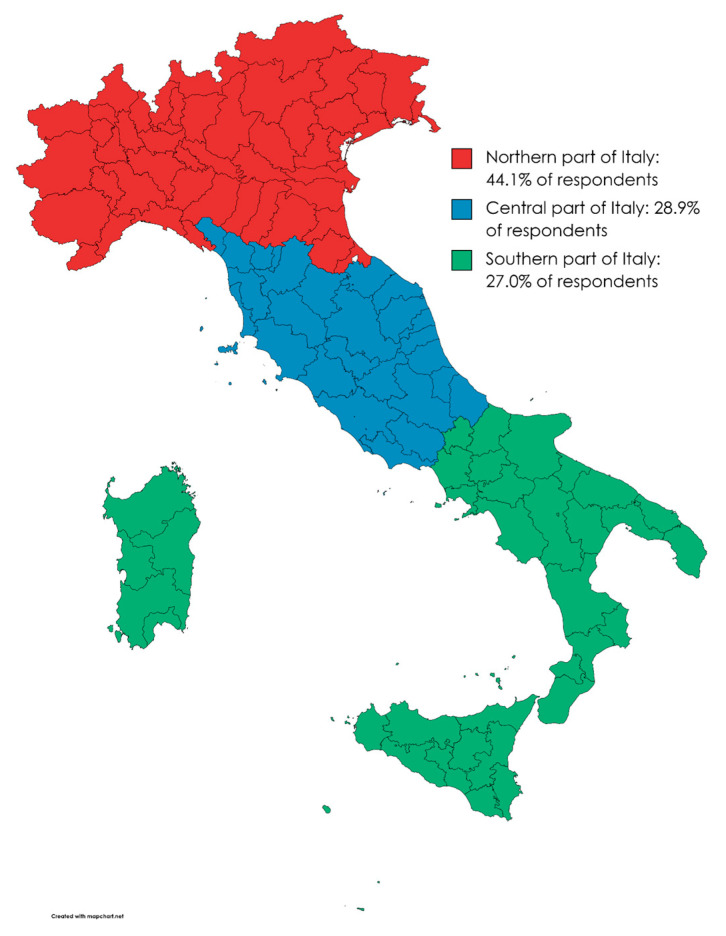
Geographical distribution of the study participants across the country.

**Table 1 healthcare-11-01557-t001:** Sociodemographic characteristics of the study sample.

	All (*n* = 1695)	Males (*n* = 239)	Females (*n* = 1456)
**Age (years. mean ± SD)**	35.3 ± 11.3	40.0 ± 13.3	34.5 ± 10.7
**Relational status**			
Engaged	1107 (65.3%)	158 (66.1%)	949 (65.2%)
Single	588 (34.7%)	81 (33.9%)	507 (34.8%)
**Have children**			
Yes	470 (27.7%)	104 (43.5%)	366 (25.1%)
1	206 (12.2%)	37 (15.5%)	169 (11.6%)
2	208 (12.3%)	54 (22.6%)	154 (10.6%)
3 or more	56 (3.4%)	13 (5.4%)	43 (3%)
No	1225 (72.3%)	135 (56.5%)	1090 (74.9%)
**Education**			
Primary School	50 (3%)	18 (7.5%)	32 (2.2%)
High School	606 (35.8%)	92 (38.5%)	514 (35.3%)
University	778 (45.9%)	103 (43.1%)	675 (46.4%)
Postgraduate	261 (15.4%)	26 (10.9%)	235 (16.1%)
**Annual income**			
<EUR 21,000 per year	835 (49.3%)	73 (30.5%)	762 (52.3%)
>EUR 21,000 per year and <EUR 60,000 per year	751 (44.3%)	128 (53.6%)	623 (42.8%)
>EUR 60,000 per year	109 (6.4%)	38 (15.9%)	71 (4.9%)

**Table 2 healthcare-11-01557-t002:** Results of the propensity-score-matched analyses. NA = not applicable; * if statistically significant.

TOTAL SAMPLE (*n* = 1695)								
Condition	Best Covariate Balance	Matched	Unmatched	Outcome	Estimated Mean Effect in the Individual with the Condition	95% CI	*p* Value	Clinically Meaningful Difference (CMD) Likelihood
**Being Male vs. Female**	Std. Mean Difference < 0.1	Controls = 1456 Cases = 239	Controls = 0 Cases = 0	Total Happiness	+3.2	+0.8 to +5.5	0.008 *	possible
				Life Perspective	0.0	−0.7 to +0.7	0.94	unlikely
				Psychophysics Status	+1.6	+1.0 to +2.2	<0.0001 *	likely
				Socio-Relational Sphere	−0.1	−0.5 to +0.4	0.77	unlikely
				Relational Private Sphere	−0.1	−0.9 to +0.7	0.75	unlikely
				Financial Status	+1.7	+0.9 to +2.5	<0.0001 *	possible
**Having an Annual Income < EUR 21,000**	Std. Mean Difference < 0.1	Controls = 860 Cases = 835	Controls = 0 Cases = 0	Total Happiness	−7.2	−9.4 to −4.9	<0.0001 *	likely
				Life Perspective	−0.6	−1.2 to 0.1	0.08	possible
				Psychophysics Status	−1.0	−1.6 to −0.4	0.002 *	possible
				Socio-Relational Sphere	−0.5	−0.9 to −0.0	0.04 *	possible
				Relational Private Sphere	−1.6	−2.2 to −1.0	<0.0001 *	ikely
				Financial Status	−3.6	−4.4 to −2.7	<0.0001 *	likely
**Having Children**	Std. Mean Difference < 0.15	Controls = 1225 Cases = 470	Controls = 0 Cases = 0	Total Happiness	−2.0	−4.6 to +0.6	0.13	possible
				Life Perspective	+0.2	−0.6 to +0.9	0.62	unlikely
				Psychophysics Status	−0.6	−1.3 to +0.2	0.12	possible
				Socio-Relational Sphere	−0.3	−0.7 to +0.2	0.22	unlikely
				Relational Private Sphere	−0.7	−1.4 to 0.1	0.07	possible
				Financial Status	−0.7	−1.9 to +0.5	0.26	possible
**Having a higher education degree**	Std. Mean Difference < 0.1	Controls = 656 Cases = 1039	Controls = 0 Cases = 0	Total Happiness	−0.3	−2.6 to +1.9	0.79	possible
				Life Perspective	+0.5	0.0 to 1.0	0.04 *	possible
				Psychophysics Status	0.0	−0.8 to +0.8	0.99	unlikely
				Socio-Relational Sphere	+0.4	−0.3 to +1.0	0.24	possible
				Relational Private Sphere	−1.3	−2.2 to −0.4	0.004 *	possible
				Financial Status	+0.1	−1–1 to +1.4	0.85	possible
**Being Engaged vs. Single**	Std. Mean Difference < 0.1	Controls = 588 Cases = 1107	Controls = 0 Cases = 0	Total Happiness	+3.0	+0.1 to +5.9	0.04 *	possible
				Life Perspective	+0.4	−0.5 to +1.2	0.41	possible
				Psychophysics Status	+0.4	−0.3 to +1.1	0.24	possible
				Socio-Relational Sphere	+0.2	−0.4 to +0.7	0.57	unlikely
				Relational Private Sphere	+1.2	+0.4 to +2.0	0.004 *	possible
				Financial Status	+1.0	+0.2 to +1.7	0.01 *	possible
**Being under 30 yo**	Std. Mean Difference < 0.1	Controls = 971 Cases = 724	Controls = 0 Cases = 0	Total Happiness	−1.0	−3.5 to +1.5	0.44	possible
				Life Perspective	+0.4	−0.2 to +1.1	0.21	possible
				Psychophysics Status	−0.6	−1.3 to +0.1	0.09	possible
				Socio-Relational Sphere	−0.3	−0.7 to +0.2	0.22	unlikely
				Relational Private Sphere	+0.4	−0.4 to +1.1	0.33	possible
				Financial Status	−0.9	−1.7 to −0.2	0.02 *	possible
**FEMALES (*n* = 1456)**								
**Having an Annual Income < EUR 21,000**	Std. Mean Difference < 0.1	Controls = 694 Cases = 762	Controls = 0 Cases = 0	Total Happiness	−7.3	−9.4 to −5.1	<0.0001 *	likely
				Life Perspective	−1.0	−1.6 to −0.4	0.001 *	possible
				Psychophysics Status	−0.9	−1.6 to −0.2	0.01 *	possible
				Socio-Relational Sphere	−0.6	−1.1 to −0.2	0.004 *	possible
				Relational Private Sphere	−1.2	−2.0 to −0.4	0.003 *	possible
				Financial Status	−3.5	−4.5 to −2.6	<0.0001 *	likely
**Having Children**	Std. Mean Difference < 0.15	Controls = 1090 Cases = 366	Controls = 0 Cases = 0	Total Happiness	−0.2	−2.9 to +2.4	0.87	possible
				Life Perspective	+1.0	+0.2 to +1.7	0.01 *	possible
				Psychophysics Status	−0.1	−0.8 to +0.6	0.83	unlikely
				Socio-Relational Sphere	−0.1	−0.6 to +0.4	0.66	unlikely
				Relational Private Sphere	−0.6	−1.4 to +0.2	0.15	possible
				Financial Status	−0.4	−1.5 to +0.7	0.47	possible
**Having a higher education degree**	Std. Mean Difference < 0.1	Controls = 546 Cases = 910	Controls = 0 Cases = 0	Total Happiness	+2.3	−0.4 to +5.1	0.1	possible
				Life Perspective	+0.6	−0.1 to +1.3	0.07	possible
				Psychophysics Status	+0.6	−0.2 to +1.3	0.15	possible
				Socio-Relational Sphere	+0.4	−0.1 to +0.9	0.09	unlikely
				Relational Private Sphere	+0.2	−0.6 to +1.0	0.60	possible
				Financial Status	+0.5	−0.4 to +1.4	0.28	possible
**Being Engaged vs. Single**	Std. Mean Difference < 0.1	Controls = 507 Cases = 949	Controls = 0 Cases = 0	Total Happiness	−0.3	−2.0 to +2.6	0.79	possible
				Life Perspective	−0.3	−0.9 to +0.3	0.29	unlikely
				Psychophysics Status	−0.4	−1.1 to +0.3	0.30	possible
				Socio-Relational Sphere	−0.3	−0.6 to +0.1	0.15	unlikely
				Relational Private Sphere	+0.9	+0.1 to +1.7	0.02 *	possible
				Financial Status	+0.4	−0.6 to +1.3	0.47	possible
**Being under 30 yo**	Std. Mean Difference < 0.1	Controls = 802 Cases = 654	Controls = 0 Cases = 0	Total Happiness	−1.4	−3.3 to +0.4	0.13	possible
				Life Perspective	+0.4	−0.3 to +1.0	0.26	possible
				Psychophysics Status	−1.3	−2.2 to −0.4	0.007 *	possible
				Socio-Relational Sphere	−0.6	−0.9 to −0.2	0.004 *	unlikely
				Relational Private Sphere	−0.3	−1.6 to +1.0	0.66	possible
				Financial Status	+0.3	−1.3 to +1.9	0.69	possible
**MALES (*n* = 239)**								
**Having an Annual Income < EUR 21,000**	Std. Mean Difference < 0.1	Controls = 166 Cases = 73	Controls = 0 Cases = 0	Total Happiness	−6.5	−10.7 to −2.3	0.002 *	likely
				Life Perspective	−0.1	−1.7 to +1.6	0.95	possible
				Psychophysics Status	−0.7	−1.8 to +0.4	0.22	possible
				Socio-Relational Sphere	−0.7	−1.8 to +0.3	0.16	possible
				Relational Private Sphere	−1.7	−3.1 to −0.3	0.02 *	possible
				Financial Status	−3.3	−5.1 to −1.5	0.0003 *	likely
**Having Children**	Std. Mean Difference < 0.15	Controls = 135 Cases = 104	Controls = 0 Cases = 0	Total Happiness	−5.7	−9.6 to −1.8	0.005 *	likely
				Life Perspective	−0.6	−2.0 to +0.8	0.40	possible
				Psychophysics Status	−1.2	−1.9 to −0.5	0.0003 *	possible
				Socio-Relational Sphere	−0.9	−1.6 to −0.2	0.01 *	possible
				Relational Private Sphere	−0.5	−1.9 to +1.0	0.52	possible
				Financial Status	−2.5	−3.6 to −1.5	<0.0001 *	likely
**Having a higher education degree**	Std. Mean Difference < 0.1	Controls = 110 Cases = 129	Controls = 0 Cases = 0	Total Happiness	+0.2	−4.7 to +5.1	0.94	possible
				Life Perspective	+0.3	−1.1 to +1.8	0.64	possible
				Psychophysics Status	+0.1	−1.1 to +1.4	0.83	possible
				Socio-Relational Sphere	−0.4	−1.2 to +0.3	0.29	possible
				Relational Private Sphere	−0.4	−1.9 to +1.1	0.58	possible
				Financial Status	+0.6	−1.2 to +2.3	0.53	possible
**Being Engaged vs. Single**	Std. Mean Difference < 0.2	Controls = 43 Cases = 158	Controls = 38 Cases = 0	Total Happiness	NA	NA	NA	
				Life Perspective	NA	NA	NA	
				Psychophysics Status	NA	NA	NA	
				Socio-Relational Sphere	NA	NA	NA	
				Relational Private Sphere	NA	NA	NA	
				Financial Status	NA	NA	NA	
**Being under 30 yo**	Std. Mean Difference < 0.1	Controls = 169 Cases = 70	Controls = 0 Cases = 0	Total Happiness	−2.6	−7.9 to +2.7	0.34	possible
				Life Perspective	−0.2	−1.7 to +1.3	0.83	possible
				Psychophysics Status	−1.3	−2.9 to +0.2	0.09	possible
				Socio-Relational Sphere	−0.3	−1.5 to +0.9	0.66	possible
				Relational Private Sphere	+0.2	−1.4 to +1.8	0.79	possible
				Financial Status	−1.1	−2.5 to +0.3	0.14	possible

## Data Availability

Data are available on request to the corresponding author.

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
