# Peer review of "Happiness and Socio-Demographic Factors in an Italian Sample: A Propensity-Matched Study"

_healthcare, 2023, doi:10.3390/healthcare11111557_

Round 1

Reviewer 1 Report

I reviewed the draft with great interest entitled “Happiness and socio-demographic factors in an Italian sample: a propensity-matched study”. The paper deals with an interesting side of the research. I have a few comments for the authors:

1.       The study lacks literature review and theoretical discussion. Which underpinning theory supports the current model? Use the following articles to develop these sections: 10.33997/j.afs.2023.36.1.004, 10.1109/DASA53625.2021.9682365

2.       Add the figure of the studied area and also discuss the demographics of the studied area.

3.       Table 1. Sociodemographic characteristics of the study sample will be more suitable in the results section.

4.       The draft is lacking data collection procedure. Add the data collection procedure and sampling techniques applied for data collection.

5.       Table 2. The results of the propensity score-matched analyses have not been interpreted. Authors should interpret the results of Table 2 before the discussion section.

6.       What are the practical and theoretical implications of the study? It is advised to add this section before the conclusion section.

7.       Also, add the limitations and future recommendations in a separate section.

Minor editing of English language required

Author Response

I reviewed the draft with great interest entitled “Happiness and socio-demographic factors in an Italian sample: a propensity-matched study”. The paper deals with an interesting side of the research. I have a few comments for the authors:

A: We thank the referee for the interest shown in our work.

  1.       The study lacks literature review and theoretical discussion. Which underpinning theory supports the current model? Use the following articles to develop these sections: 10.33997/j.afs.2023.36.1.004, 10.1109/DASA53625.2021.9682365

A: We have followed this recommendation and further developed a theoretical discussion enriched with a literature review in the Introduction section of the manuscript. We have cited the two papers proposed by the referee, which showed to be of interest for the theoretical background.

  1.       Add the figure of the studied area and also discuss the demographics of the studied area.

A: A figure with the geographical distribution of study respondents has been added to the manuscript. The most relevant socio-demographic characteristics of the Italian population have been reported in the article (see the Results section).

  1.       Table 1. Sociodemographic characteristics of the study sample will be more suitable in the results section.

A: Sociodemographic characteristics of the study sample have been moved to the Results section of the manuscript.

  1.       The draft is lacking data collection procedure. Add the data collection procedure and sampling techniques applied for data collection.

A: Data collection procedure has been better described in the Methods section.

  1.       Table 2. The results of the propensity score-matched analyses have not been interpreted. Authors should interpret the results of Table 2 before the discussion section.

A: The Results section has been expanded and further details have been reported to help the reader understand our findings in more depth. The results of our survey are interpreted in the Discussion section.

  1.       What are the practical and theoretical implications of the study? It is advised to add this section before the conclusion section.

A: Implications of the study have been detailed in the Conclusions section.

  1.       Also, add the limitations and future recommendations in a separate section.

A: The Limitations section has been expanded and better described. Another section (entitled “Recommendations and implications of the study”) has been added to the article. 

Reviewer 2 Report

Dear Editors

Dear authors,

Thank you for the opportunity to read and review this paper. this study assesses how the levels of happiness vary in a large sample of Italian adults and to identify the sociodemographic conditions which impair happiness domains the most.

I really enjoy to read this articles and the findings is really interesting that Having children appears to have a negative impact on male happiness.

Before considering publishing this paper, this paper needs several improvements.

I believe that this study need literature review or theoretical framework section after introduction section. Here, I have read some interesting articles about happiness and well-being.

T. T. Wijaya, I. F. Rahmadi, S. Chotimah, J. Jailani, and D. U. Wutsqa, “A Case Study of Factors That Affect Secondary School Mathematics Achievement: Teacher-Parent Support, Stress Levels, and Students’ Well-Being,” Int. J. Environ. Res. Public Health, vol. 19, no. 23, 2022, doi: 10.3390/ijerph192316247.

M. Salanova, W. Schaufeli, I. Martínez, and E. Bresó, “How obstacles and facilitators predict academic performance: The mediating role of study burnout and engagement,” Anxiety, Stress Coping, vol. 23, no. 1, pp. 53–70, 2009, doi: 10.1080/10615800802609965.

More than 80 percent of the respondents in this study were women. I doubt that there will be research bias here. How can the author generalize the results of this study?

Please add the questionnaire at appendix section or in methodology section. It will be a valuable information for reader and other researchers.

Data analysis: what is the step to analysis the data with R need to explain clearly.

In discussion section author need to remind the reader what is the problem and how they solve the problem.

Please add the recommendation sections and implication sections. What are the implications and recommendations of this research for society, government and for increasing happiness and well-being?

Author Response

Dear authors,

Thank you for the opportunity to read and review this paper. this study assesses how the levels of happiness vary in a large sample of Italian adults and to identify the sociodemographic conditions which impair happiness domains the most.

I really enjoy to read this articles and the findings is really interesting that Having children appears to have a negative impact on male happiness.

A: We thank the referee for the interest shown in our work.

Before considering publishing this paper, this paper needs several improvements.

I believe that this study need literature review or theoretical framework section after introduction section. Here, I have read some interesting articles about happiness and well-being.

  1. T. Wijaya, I. F. Rahmadi, S. Chotimah, J. Jailani, and D. U. Wutsqa, “A Case Study of Factors That Affect Secondary School Mathematics Achievement: Teacher-Parent Support, Stress Levels, and Students’ Well-Being,” Int. J. Environ. Res. Public Health, vol. 19, no. 23, 2022, doi: 10.3390/ijerph192316247.
  2. Salanova, W. Schaufeli, I. Martínez, and E. Bresó, “How obstacles and facilitators predict academic performance: The mediating role of study burnout and engagement,” Anxiety, Stress Coping, vol. 23, no. 1, pp. 53–70, 2009, doi: 10.1080/10615800802609965.

A: A literature review has been added in the Introduction section. The papers mentioned have been cited in the article.

More than 80 percent of the respondents in this study were women. I doubt that there will be research bias here. How can the author generalize the results of this study?

A: The over-representation of female participants in the study sample has been acknowledged in the Limitations section. Moreover, the number of males was still quite high (n=239), sufficient to reach a good statistical power also in post-hoc analysis (MH is a continuous endpoint) with a 99,4% power to detect a difference between two different groups with an alpha set at 0.05, Finally, the propensity-matched analysis was used to allow balanced gender-based groups that, although numerically different, had adequately matched characteristics, even if the matchings followed ratios superior to 1:1.

Please add the questionnaire at appendix section or in methodology section. It will be a valuable information for reader and other researchers.

A: As requested by the referee, the MH questionnaire has been added in the method section of the article.

Data analysis: what is the step to analysis the data with R need to explain clearly.

A: The steps needed for the propensity matching analysis are described in the methods section (2.3 Data Analysis) following exactly the reporting rules as indicated by the official guide of the R package MatchIt. However, additional references on the topic have been cited. 

In discussion section author need to remind the reader what is the problem and how they solve the problem.

A: A brief summary of the study objectives and findings have been reported in the first part of the Discussion section. 

Please add the recommendation sections and implication sections. What are the implications and recommendations of this research for society, government and for increasing happiness and well-being?

A: Recommendations and implications of the study have been detailed in the Conclusions section.

Reviewer 3 Report

We appreciate the clarity in the presentation of the results: financial aspects are important and have an impact on the other aspects, and sharing life with a partner is also relevant. The ambivalence about having children is also interesting.
On the other hand, the references to the political sphere in order to strengthen the economic and affective aspects, as well as to seek strategies in relation to having children, are welcome.
It is right to point out the limitation of the research, although it would be appreciated not to focus this limitation only on the age ranges, but also on the size of the sample in relation to the population and the low percentage of men as opposed to women.
It is possible that very similar results would be found in many neighbouring countries.

Author Response

A: We would like to thank the referee for the appreciation of our research. We have followed the recommendation to better detail the study limitations. We have now mentioned that future studies should include a sample as large as possible and a population characterized by a more balanced sex ratio, and included post-hoc power analysis showing that the male sample is quite sufficient.

Reviewer 4 Report

I was pleased to get acquainted with the content of the article "Happiness and socio-demographic factors in an Italian sample: a propensity-matched study?" submitted for review (a team of authors - Matteo Rizzato, Michele Antonelli, Carlo Sam, Cinzia Di Dio, Davide Lazzeroni, Davide Donelli). The research problem stated by the authors is relevant and timely, both for a narrow circle of specialists and for a wide audience. The novelty of the research lies in the study of socio-demographic factors and the degree of their influence on happiness (the experience of happiness). The results of the study enrich the knowledge about the impact of socio-demographic factors on the quality and standard of living of the population. The results of the study can be taken into account directly by the residents themselves in shaping the trajectory of their life, as well as in assisting the relevant categories of citizens in building their life path.

Research on such topics is relevant and in demand in the practice of healthcare organizations. Literary sources and references to them are used appropriately.

I would like, however, to see in the list of sources the works of Petretto R.D., Pili R. (Italy, Cagliari), carrying out a scientific search in a similar direction.

But the absence of references to the works of these authors in no case can be an obstacle to the publication of the article.

Author Response

A: We would like to thank the referee for the appreciation of our research. This article by Petretto and Pili has been referenced in our study (10.3390/geriatrics5020025).

Round 2

Reviewer 1 Report

Authors followed the comments to revise the draft and current draft is more reader friendly. 

No comment about language

Reviewer 2 Report

author has been revised the manuscript very well.

now, the manuscript ready to publish.

well Done.